# PeerJ

# Succession of the turkey gastrointestinal bacterial microbiome related to weight gain

Jessica L. Danzeisen[1], Alamanda J. Calvert[2], Sally L. Noll[2], Brian McComb[3], Julie S. Sherwood[4], Catherine M. Logue[4,5] and Timothy J. Johnson[1]

[1] Department of Veterinary and Biomedical Sciences, College of Veterinary Medicine, University of Minnesota, Saint Paul, MN, USA
[2] Department of Animal Sciences, College of Food, Agriculture and Natural Resource Sciences, University of Minnesota, Saint Paul, MN, USA
[3] Willmar Poultry Company, Willmar, MN, USA
[4] Department of Veterinary and Microbiological Sciences, North Dakota State University, Fargo, ND, USA
[5] Department of Veterinary Microbiology and Preventive Medicine, College of Veterinary Medicine, Iowa State University, Ames, IA, USA

Corresponding author
Timothy J. Johnson,
joh04207@umn.edu

## ABSTRACT

Because of concerns related to the use of antibiotics in animal agriculture, antibiotic-free alternatives are greatly needed to prevent disease and promote animal growth. One of the current challenges facing commercial turkey production in Minnesota is difficulty obtaining flock average weights typical of the industry standard, and this condition has been coined "Light Turkey Syndrome" or LTS. This condition has been identified in Minnesota turkey flocks for at least five years, and it has been observed that average flock body weights never approach their genetic potential. However, a single causative agent responsible for these weight reductions has not been identified despite numerous efforts to do so. The purpose of this study was to identify the bacterial community composition within the small intestines of heavy and light turkey flocks using 16S rRNA sequencing, and to identify possible correlations between microbiome and average flock weight. This study also sought to define the temporal succession of bacteria occurring in the turkey ileum. Based upon 2.7 million sequences across nine different turkey flocks, dominant operational taxonomic units (OTUs) were identified and compared between the flocks studied. OTUs that were associated with heavier weight flocks included those with similarity to Candidatus division Arthromitus and *Clostridium bartlettii*, while these flocks had decreased counts of several *Lactobacillus* species compared to lighter weight flocks. The core bacterial microbiome succession in commercial turkeys was also defined. Several defining markers of microbiome succession were identified, including the presence or abundance of Candidatus division Arthromitus, *Lactobacillus aviarius*, *Lactobacillus ingluviei*, *Lactobacillus salivarius*, and *Clostridium bartlettii*. Overall, the succession of the ileum bacterial microbiome in commercial turkeys proceeds in a predictable manner. Efforts to prevent disease and promote growth in the absence of antibiotics could involve target dominant bacteria identified in the turkey ileum that are associated with increased weight gain.

## INTRODUCTION

The United States produces more than 250 million turkeys per year (*USDA, 2012*), resulting in a multi-billion dollar per year industry (*USDA, 2007*). Growth performance and sustained flock health is of major economic importance to commercial turkey producers. The microbial community of the gastrointestinal tract, or microbiome, is assumed to play a critical role in overall health of turkeys and other poultry, but the composition of bacterial species within the turkey gastrointestinal tract is largely understudied. Furthermore, the role of these microbes in the development of a healthy intestinal tract is not entirely understood.

The onset of next-generation sequencing has resulted in a marked increase in culture-independent studies characterizing the gut microbiome, but much of this work has been focused on humans and other production animals (*Danzeisen et al., 2011*; *Kim et al., 2011*; *Scalzo et al., 2004*). Fewer studies have sought to understand the turkey microbiome. Some work has focused on comparison of the cecal microbiomes of wild and domestic birds (*Scupham et al., 2008*), or examination of the turkey microbiome in relation to pathogen colonization, such as *Campylobacter* (*Scupham, 2009*). These studies identified specific differences in the genera present in the ceca of differing types of turkeys, as well as time-dependent shifts in bacterial populations in the turkey intestinal tract. However, such previous turkey studies relied on culture-based methods or lower output molecular fingerprinting methods, such as T-RFLP (*Scupham, 2009*; *Scupham et al., 2008*). These methods present limitations and biases that are not encountered with high-throughput sequencing (*Handelsman, 2004*), which can be targeted at loci such as the 16S rRNA gene to achieve deep taxonomic coverage and better resolution.

Commercial turkey production in Minnesota is currently facing a widespread challenge in obtaining flock average weights typical of the industry standard, and this condition has been coined "Light Turkey Syndrome" or LTS (personal communication with the Minnesota Turkey Growers Association). Lower than expected market weights associated with LTS seem to be most prevalent in heavy tom flocks, since the problem is likely amplified over the longer growth period of toms versus hens (*Calvert, 2012*). LTS has been identified in Minnesota turkey flocks for at least five years, and it has been observed that average flock body weights never approach their genetic potential. The gap between observed weights and genetic potential starts immediately at brooding and continues throughout the grow-out phase of the turkey (*Calvert, 2012*). A number of possible contributing factors have been speculated, including management practices, the presence of known or unknown bacterial or viral pathogens, disruptions of the gastrointestinal microbial communities, problems with nutrient absorption, or dwarfed immune development in poults (*Calvert, 2012*). This problem has not, to our knowledge, been identified in other states in the USA.

It is unclear if LTS is tied to another condition in turkeys called poult enteritis syndrome, or PES (*Jindal et al., 2009*). Studies have demonstrated that inoculation of healthy birds with fecal slurries from birds experiencing PES results in significant reductions in body weight compared to controls (*Mor et al., 2011*). However, a single causative agent responsible for these weight reductions has not been identified. Astroviruses, rotaviruses, and reoviruses have been found in birds experiencing PES, but they are also found in apparently healthy birds (*Jindal et al., 2012*) or studies examining these viruses lacked negative control groups (*Jindal et al., 2010*). LTS has been reproduced in controlled animal studies using pooled fecal homogenates, but not individual microorganisms (*Mor et al., 2013*). Also, no epidemiological associations have been made between the PES and LTS, and LTS often occurs in the absence of PES. Overall, there is no evidence to suggest a single pathogen associated with LTS.

The use of subtherapeutic concentrations of antimicrobial agents in animal agriculture (i.e., growth promoters) is being increasingly scrutinized because of the rise of multidrug resistant pathogens. Therefore, there is an urgent need to identify antibiotic-free alternatives to improving animal health and weight gain. If there is a microbiome association with conditions such as LTS in turkeys, it is therefore necessary to better understand the succession of bacterial communities in the gastrointestinal tract as the bird ages prior to successfully modulating these microbes using antibiotic-free approaches. Thus, the purpose of this study was to examine bacterial community succession in turkeys raised under industry conditions, and to compare the bacterial communities of young turkeys of different average flock weights using 16S rRNA microbiome analysis.

## MATERIALS AND METHODS

### Sample collection

All animal experiments were performed in accordance with the Institutional Animal Care and Use Committee at the University of Minnesota, protocol 1012B93592. Two independent experiments were performed. In experiment #1, we examined flock-level differences in the ileum bacterial microbiome using pooled samples from multiple turkey flocks of differing average daily weight. Five commercial flocks (designated flocks #1–2, #4, and #7–8) from three different facilities were examined, and two research flocks (designated flocks #5–6) were examined (Table 1). At three time points (1, 2, and 3 weeks of age, +/− 3 days), 20–40 birds were selected per flock/timepoint and humanely euthanized using AVMA approved methods. Ileal sections of the small intestine were aseptically collected, homogenized and immediately frozen. Homogenates included both intestinal content and intestinal wall to assess the total bacterial content in the ileum. Sample poult weights for each flock were determined at each collection timepoint.

In experiment #2, two commercial turkey flocks were followed temporally from 1 week to 12 weeks of age (Table 1). From weeks 1–6 of age, birds were sampled at weekly timepoints, followed by sampling every two weeks through 11 or 12 weeks of age. Individual birds were euthanized and their ileum contents collected as described above. For each euthanized bird, total bird weights, intestinal weights, and intestinal lengths were recorded.

**Table 1 Samples analyzed in this study.** First letter in sample refers to flock sampled, with F, Light/Heavy flocks; CF, commercial flock individual samples; and RF, research flock individual samples; "W", refers to week of age.

| Sample | Flock type | Birds ($n$) | Total filtered sequences | Total rarefied sequences | Description |
|---|---|---|---|---|---|
| F1W1 | Commercial | 40 | 54,868 | 20,000 | Pooled Light commercial flock |
| F1W2 | Commercial | 40 | 96,067 | 20,000 | Pooled Light commercial flock |
| F1W3 | Commercial | 40 | 46,694 | 20,000 | Pooled Light commercial flock |
| F2W1 | Commercial | 40 | 35,379 | 20,000 | Pooled Light commercial flock |
| F2W2 | Commercial | 40 | 42,313 | 20,000 | Pooled Light commercial flock |
| F2W3 | Commercial | 40 | 43,083 | 20,000 | Pooled Light commercial flock |
| F4W1 | Commercial | 40 | 19,651 | 20,000 | Pooled Light commercial flock |
| F4W2 | Commercial | 20 | 20,472 | 20,000 | Pooled Light commercial flock |
| F4W3 | Commercial | 40 | 47,365 | 20,000 | Pooled Light commercial flock |
| F5W1 | Research | 40 | 62,525 | 20,000 | Pooled Heavy research flock |
| F5W2 | Research | 20 | 7,870 | 7,870 | Pooled Heavy research flock |
| F5W3 | Research | 20 | 21,258 | 20,000 | Pooled Heavy research flock |
| F6W1 | Research | 40 | 41,978 | 20,000 | Pooled Heavy research flock |
| F6W2 | Research | 40 | 56,625 | 20,000 | Pooled Heavy research flock |
| F6W3 | Research | 40 | 42,878 | 20,000 | Pooled Heavy research flock |
| F7W1 | Commercial | 20 | 19,877 | 20,000 | Pooled Light commercial flock |
| F7W2 | Commercial | 40 | 39,240 | 20,000 | Pooled Light commercial flock |
| F7W3 | Commercial | 20 | 14,785 | 20,000 | Pooled Light commercial flock |
| F8W1 | Commercial | 40 | 49,918 | 20,000 | Pooled Heavy research flock |
| F8W2 | Commercial | 40 | 85,002 | 20,000 | Pooled Heavy research flock |
| F8W3 | Commercial | 20 | 8,855 | 8,855 | Pooled Heavy research flock |
| RFW1 (1–5) | Research | 5 | 365,704 | 100,000 | Individuals from research flock |
| RFW2 (1–5) | Research | 5 | 465,709 | 100,000 | Individuals from research flock |
| RFW3 (1–5) | Research | 5 | 394,010 | 100,000 | Individuals from research flock |
| RFW4 (1–5) | Research | 5 | 400,081 | 100,000 | Individuals from research flock |
| RFW5 (1–5) | Research | 5 | 387,436 | 100,000 | Individuals from research flock |
| RFW6 (1–5) | Research | 5 | 456,102 | 100,000 | Individuals from research flock |
| RFW8 (1–5) | Research | 5 | 514,068 | 100,000 | Individuals from research flock |
| RFW10 (1–5) | Research | 5 | 283,898 | 100,000 | Individuals from research flock |
| RFW12 (1–5) | Research | 5 | 2,449,503 | 100,000 | Individuals from research flock |
| CFW1 (1–10) | Commercial | 10 | 1,691,968 | 200,000 | Individuals from commercial flock |
| CFW2 (1–10) | Commercial | 10 | 1,622,891 | 200,000 | Individuals from commercial flock |
| CFW3 (1–10) | Commercial | 10 | 1,701,982 | 200,000 | Individuals from commercial flock |
| CFW4 (1–10) | Commercial | 10 | 1,247,984 | 200,000 | Individuals from commercial flock |
| CFW5 (1–10) | Commercial | 10 | 1,723,505 | 200,000 | Individuals from commercial flock |
| CFW6 (1–9) | Commercial | 9 | 1,673,269 | 180,000 | Individuals from commercial flock |
| CFW7 (1–5) | Commercial | 5 | 635,010 | 100,000 | Individuals from commercial flock |
| CFW9 (1–5) | Commercial | 5 | 766,290 | 100,000 | Individuals from commercial flock |
| CFW11 (1–5) | Commercial | 5 | 481,758 | 100,000 | Individuals from commercial flock |
| | Total | 834 | 18,117,871 | 2,776,725 | |

## DNA Extraction and 16S rRNA amplification

For experiment #1, ileum samples were pooled together according to flock, time point and weight status. For experiment #2, individuality of the samples was retained. DNA was extracted using a bead-beating procedure and the QIAmp® DNA Stool Kit (Qiagen, Valencia, CA) as previously described (*Danzeisen et al., 2011*). The V3 hypervariable region of the 16S rRNA gene was amplified in 25 µl reactions containing 1X PCR buffer (containing 1.8 mM MgCl$_2$), 0.2 mM each dNTP (Promega, Madison, WI), 0.4 µM each primer (Integrated DNA Technologies, Coralville, IA), 1.25 U FastStart High Fidelity *Taq* polymerase (Roche, Basel, Switzerland). Primers were designed for Illumina barcoding and sequencing as previously described (*Bartram et al., 2011*). Each forward and reverse primer contained a sample-specific sequence barcode. The PCR conditions used were an initial denaturation of 95°C for 2 min, followed by 25 cycles of 95°C for 30 s, 60°C for 30 s and 72°C for 30 s; the amplification was completed with a final extension of 72°C for 7 min. The PCR product was excised from a 1.5% gel and purified using the QIAquick Gel Extraction Kit following manufacturer's instructions (Qiagen). Sample DNA quality and quantity were assessed on a Bioanalyzer 2100 (Agilent, Palo Alto, CA) using a DNA-1000 lab chip. Sequencing was performed at the University of Minnesota using Illumina MiSeq paired-end 2X250 bp technology.

## Data analysis

Following sequencing, sorting by barcode was performed to generate fastq files for each sample. Paired end reads were assembled and quality screened using Pandaseq (*Masella et al., 2012*). To reduce the effects of random sequencing errors, sequences that met any of the following criteria were eliminated: sequences that did not match the PCR primers and barcode; sequences that were truncated; sequences with one or more undetermined nucleotide (N); and sequences with an average Phred score $\leq 27$. Proximal and distal primers were trimmed from the remaining sequence reads prior to database searches and similarity calculations. A de novo operational taxonomic unit (OTU) picking approach was used in QIIME (*Caporaso et al., 2010*) using uclust (*Edgar, 2010*). Potential chimeras were removed using ChimeraSlayer (*Edgar, 2010*). Approximately-maximum-likelihood phylogenetic trees were constructed using FastTree (*Price, Dehal & Arkin, 2010*). QIIME was also used for assessments of alpha diversity, beta diversity using Unifrac (*Lozupone & Knight, 2005*), and phylogenetic classifications using the RDP database (*Cole et al., 2009*; *Wang et al., 2007*). Differential abundances of OTUs and other phylogenetic classifications were identified using METASTATS (*White, Nagarajan & Pop, 2009*). Construction of heatmaps was performed using the R statistical software (*Dean & Nielsen, 2007*). MEGA5 was used for inferring the phylogenetic relationships between known *Lactobacillus* species and similar OTUs (*Tamura et al., 2011*). A cladogram depicting OTU relationships was developed using the Interactive Tree of Life (iTOL) (*Letunic & Bork, 2007*).

## Light microscopy

From 20 individual samples with known variations in abundance of an OTU similar to Candidatus division Arthromitus, 0.25 g of homogenized material was diluted in 10 ml

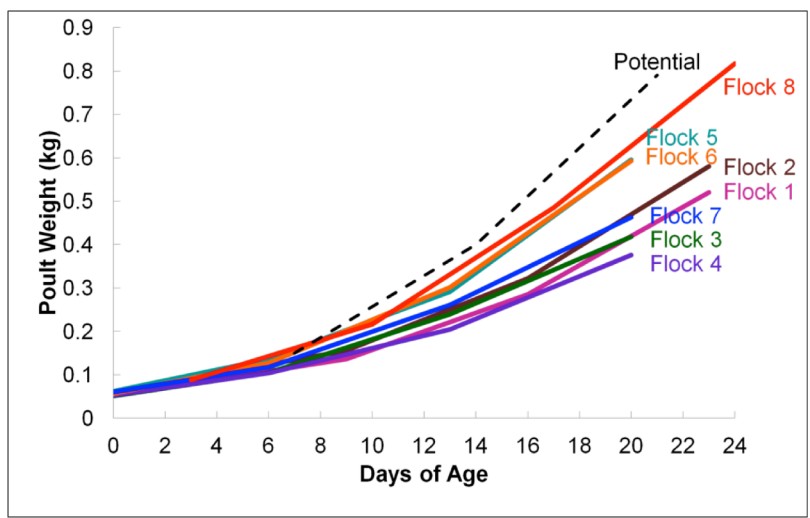

**Figure 1 Average flock weights for Light versus Heavy flocks in experiment #1.** To calculate the average sample flock weight, 40 birds from each flock were weighed or an average weight from the barn scale readings was used. Dashed line indicates genetic potential of bird type used.

phosphate buffered saline. From this sample, 10 µl was fixed onto a microscope slide and stained with crystal violet. Five fields per sample were used to count and average segmented filamentous bacteria within the field.

## RESULTS

A total of 140 samples were analyzed in this study, involving pooled samples for flock-level comparisons (experiment #1) and individual samples for within-flock comparisons (experiment #2). From these samples, a total of 18,117,871 2X250 bp Illumina MiSeq reads were generated, assembled and quality screened using Pandaseq, trimmed of primer/barcode, and rarefied to 2,776,725 sequences. These remaining sequences were analyzed using QIIME (Table 1). After removal of singleton OTUs upon clustering, there were 2,469 OTUs that remained in the dataset representing species-level OTUs across all samples at 97% clustering.

### Experiment #1: flock-level comparisons

According to average daily poult weights, flocks were classified into two groups, light (flocks #1–4 and #7) and heavy (flocks #5–6 and #8) which were significantly different from one another based upon weight ($P < 0.05$; Fig. 1). Subsequent comparisons of sequencing data were based upon these classifications. In total, 396,725 16S rRNA sequences were analyzed for flock-level comparisons (Table 1).

RDP was used to analyze 16S rRNA sequence reads at the class level, using a bootstrap confidence threshold of 50% based upon RDP recommendations for short read lengths (*Cole et al., 2009*). Using RDP analysis, Firmicutes was the most prevalent phylum in all flocks for the duration of the experiment (Fig. 2), comprised mostly of Bacilli. Gammaproteobacteria made up substantial proportions of the bacterial microbiome at week 1 of age in both the Heavy and Light flocks. Genus-level distribution showed that

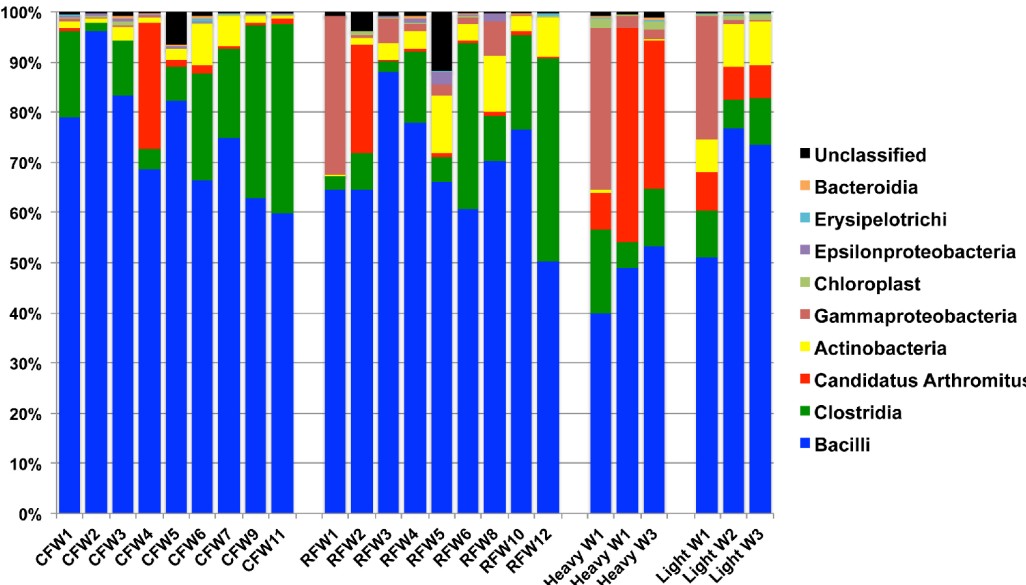

**Figure 2** **Taxonomic classification of groups in this study.** Class-level taxonomic classification was based upon average proportional abundance of normalized samples. Designations in the *X*-axis are "CF", individual birds from commercial flock (experiment #2); "RF", individual birds from research flock (experiment #2); "Heavy", pooled samples categorized as heavy weights (experiment #1); and "Light", pooled samples categorized as lighter weights (experiment #1). "W", age of birds in weeks.

these populations were largely composed of *Pseudomonas* species in flocks #1 and #5, but were mostly *Escherichia coli* in flock #8. Flocks #1 and #2 also showed a higher occurrence of the class Actinobacteria than the other five flocks, which was composed mostly of the genus *Bifidobacterium*. Overall, the RDP analysis showed on the class level that Bacilli and Actinobacteria were significantly higher ($P < 0.05$) on average in the Light flocks. The Heavy flocks had higher proportions of an unknown bacteria at weeks 2 and 3 of age, which was later identified as Candidatus division Arthromitus (see below).

Overall, the number of observed OTUs fluctuated from weeks 1 to 3 of age, with no discernible pattern in any of the 7 flocks (Table S1). This is also demonstrated through Chao1, Shannon and Simpson richness and diversity indices, again suggesting an immature and changing gut microbiome during the first three weeks of life. The rarefaction curve (Fig. S1) also suggested that there was no discernable pattern to the curves from weeks 1 to 3 of age in the flocks; if the diversity and richness was increasing over the 21 day time period of this study, one might expect the slope of the curves from weeks 1 to 3 of age to also increase accordingly. Inferred phylogeny of the collective turkey ileum microbiome based upon OTU (Fig. 3) was comparable to the taxonomic classifications performed with RDP, with most of the diversity of the microbiome being attributed to orders Clostridia and Bacilli. Unifrac-based distance matrices demonstrated that there were no significant differences in the overall microbiome compositions of Light versus Heavy flock at any of the three timepoints (Fig. S2). However, OTU-based analysis revealed significant changes in bacterial taxa, as described below.

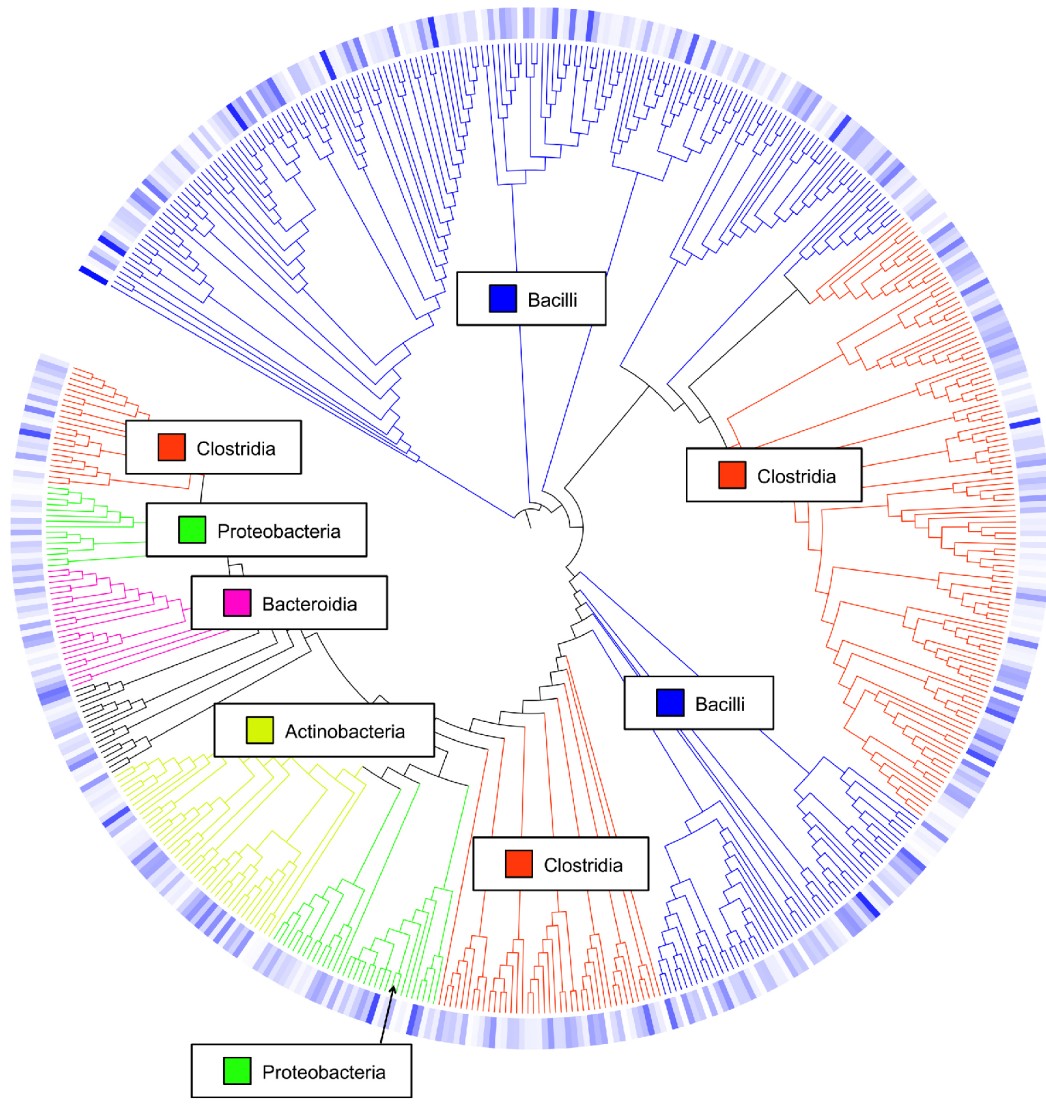

**Figure 3 Cladogram of operational taxonomic units (OTUs).** Cladogram illustrating relationships of OTUs in the entire study, classified using the Ribosomal Database Project and/or BLAST. The outer ring depicts relative $\log_{10}$ abundances of OTUs in the entire dataset. Figure was generated using the Interactive Tree of Life (iTOL).

Among the observed OTUs in the entire dataset, the ten most abundant OTUs represented 78.6% of the total sequences obtained (Table S2). These dominant OTUs were classified using RDP database and BlastN, and included representative sequences with similarity to *Lactobacillus salivarius*, *Lactobacillus acidophilus/crispatus/helveticus* (*L. delbrueckii* group), *Lactobacillus aviarius*, *Lactobacillus johnsonii*, *Clostridium bartlettii* (*Clostridium* group XI), *Lactobacillus reuteri/vaginalis*, Candidatus division Arthromitus, *Enterococcus* species, *E. coli*, and *Roseburia* species. In some cases, the V3 region was sufficient for BLAST similarity enabling classification down to the species level at 100% similarity to database sequences, as demonstrated for certain *Lactobacillus* species in

Fig. 4. The dominant OTUs were not evenly distributed among the flocks examined. In fact, a clear trend was observed where the flocks classified as Heavy contained greater proportions of OTUs classified as Candidatus division Arthromitus and *C. bartlettii*, and lesser proportions of OTUs classified as *Lactobacillus* species. METASTATS was performed to determine significant enrichments or depletions in OTUs in the flocks classified as Light versus those classified as Heavy (Table S3). We chose METASTATS because it employs a false discovery rate to deal with high-complexity environments and uses Fisher's exact test to deal with low abundance or sparsely sampled features (*White, Nagarajan & Pop, 2009*). At week 1 of age, significantly higher proportions ($P < 0.05$) of OTUs were observed in Heavy flocks with similarity to *Anaerobacter* species, *Clostridium sensu stricto*, and *Pseudomonas* species. In birds two weeks of age, most of the significant changes in Light versus Heavy flocks included higher proportions in OTUs classified as *Clostridium bartlettii* and Candidatus division Arthromitus and depletions in a number OTUs classified as *Lactobacillus* spp. in Heavy flocks. At birds three weeks of age, reduced proportions were also observed in OTUs classified as *Lactobacillus* species along with increases in the proportions of Candidatus division Arthromitus OTU and several OTUs classified as *Lactococcus* species in Heavy flocks. Light microscopy was used to validate that sequences with similarity to Candidatus division Arthromitus appeared as segmented filamentous bacteria (Fig. 5). Indeed, segmented filamentous organisms were easily identified in samples that contained large proportions of sequences with similarity to Candidatus division Arthromitus, while these organisms were absent in samples lacking sequences with similarity to Candidatus division Arthromitus. The average number of organisms identified in 5 fields per sample correlated with the counts of the OTU similar to Candidatus division Arthromitus ($R^2 = 0.92$).

## Experiment #2: within-flock comparisons

In experiment #2, two flocks were followed from ages 1–12 weeks and individual ileum bacterial microbiomes were compared. A total of 119 individual turkey ileum samples were collected from multiple birds at each timepoint examined, resulting in 2,380,000 sequences that were analyzed in QIIME. The dominant OTUs identified in experiment #2 were the same dominant OTUs identified in experiment #1. However, following individual birds over extended periods of time enabled better resolution related to the succession of bacterial populations in the turkey ileum. The flocks followed included a typical commercial turkey flock (designated CF) of 28,000 hens and a research flock (designated RF) replicating commercial farm conditions. Both flocks were sampled at approximately the same timeframe, used the same hatchery as a source of poults, and used similar nutritional plans. The two flocks were strikingly similar in their ileum bacterial microbiome succession, with no significant differences at community-level comparisons based on average Unifrac distances at the same age timepoints (Fig. S3). An average heatmap was constructed to visualize OTU averages for each flock at each timepoint (Fig. 6). In both flocks, birds possessed high proportions of OTUs with similarity to *L. salivarius*, *L. reuteri/vaginalis*, and *L. acidophilus/crispatus/helveticus* (*L. delbrueckii*

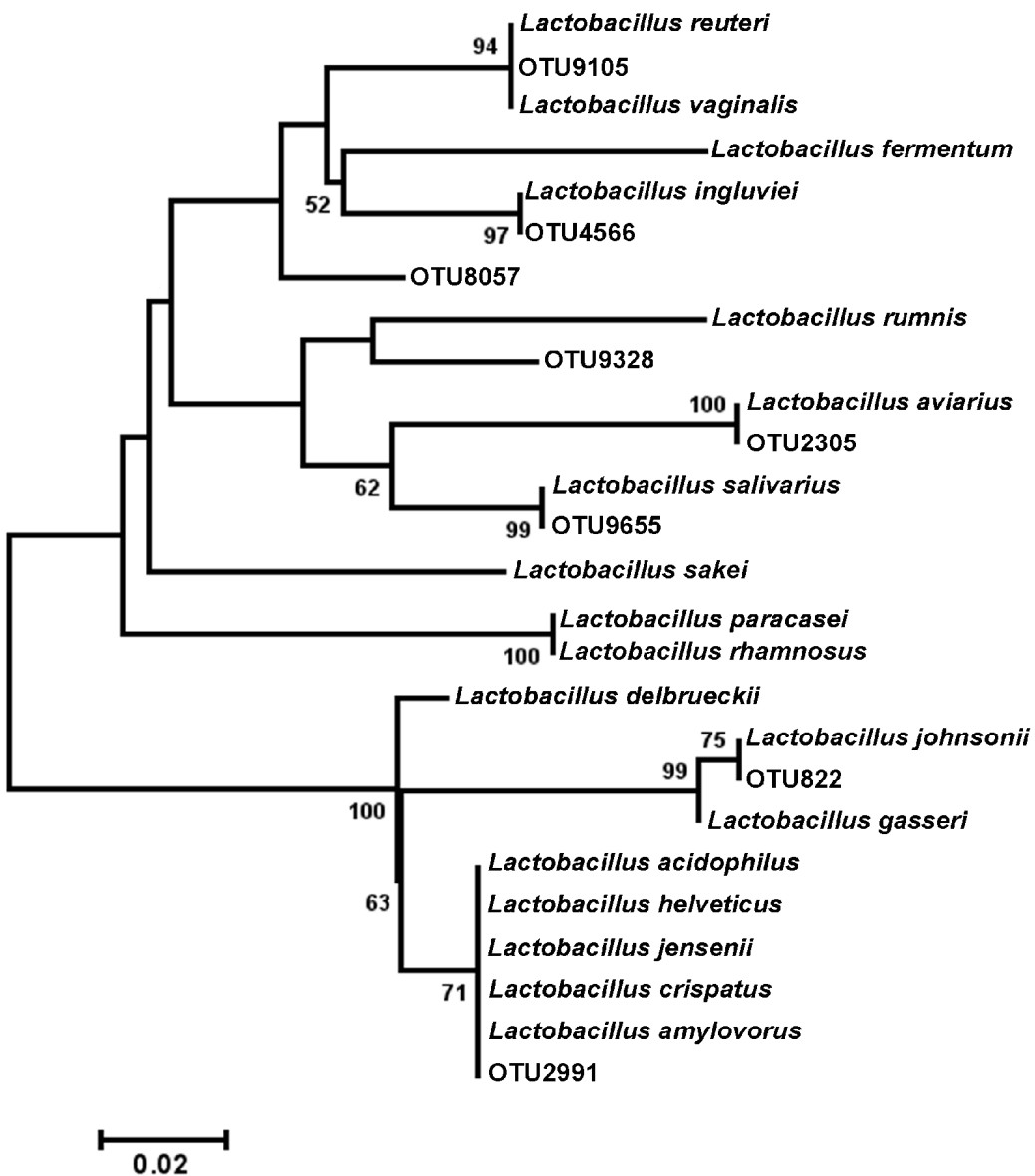

**Figure 4 Dendrogram of operational taxonomic units (OTUs) with similarity to *Lactobacillus*.** Phylogenetic relationships were inferred using Maximum Likelihood analysis with a General Time Reversible Model using 1,000 bootstrap replicates in MEGA5. The dataset was generated from representative OTUs with similarity to *Lactobacillus* spp., and extracted *Lactobacillus* sequences from the NCBI database for the V3 hypervariable region of the 16S rRNA region.

group) throughout the 12 weeks of the study. In contrast, OTUs with similarity to *L. aviarius* and *L. johnsonii* appeared at 2–4 weeks of age and subsequently increased or maintained their proportions with age in both flocks. An OTU with similarity to *C. bartlettii* appeared in both flocks at week 5 of age. A number of lower abundance OTUs also appeared over time in a predictable manner in both flocks studied. One key difference between the two flocks studied was the timing of the appearance of the OTU with similarity

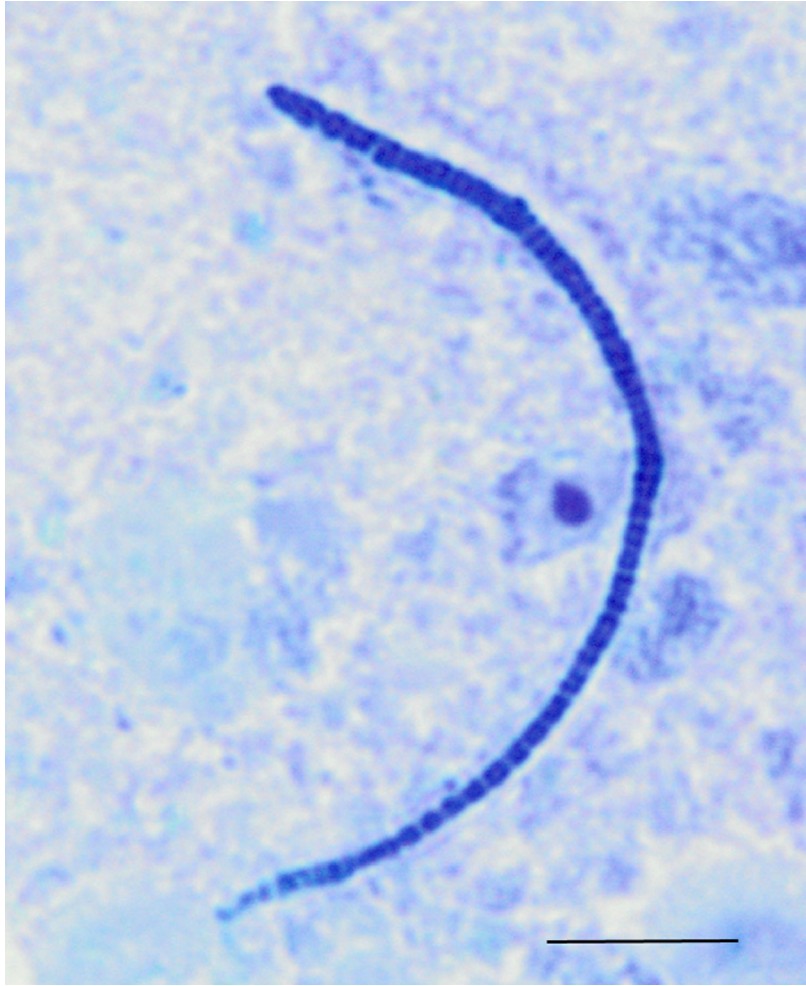

**Figure 5 Light microscopy of segmented filamentous bacteria in the turkey ileum.** Scaled line is approximately 10 microns.

to Candidatus division Arthromitus, increasing in abundance at 4 weeks of age in the commercial flock and at 2 weeks of age in the research flock. While the proportions of some OTUs varied in individual birds of the same flock and timepoint, overall there were discernable patterns and consistency in the temporal succession of the bacterial microbiome in individual birds (Fig. 7). Sample diversity was assessed temporally using the Shannon index as a measure of community diversity and Chao1 as an estimator of community richness (Fig. S4), and this demonstrated that diversity and richness increased temporally until 8 weeks of age where it plateaued.

Principal coordinate analysis (PCoA) of all samples in the study clearly demonstrated that bird age was more influential than flock or Heavy/Light classification (Fig. 8). Also evident was that as turkey age increased, the individual birds shared greater bacterial community similarity, particularly after the move of birds from brood to grow-out. PCoA coordinate PC2 was plotted against individual bird whole bird weights, and demonstrated that there was a predictable shift in the bacterial community from hatch through
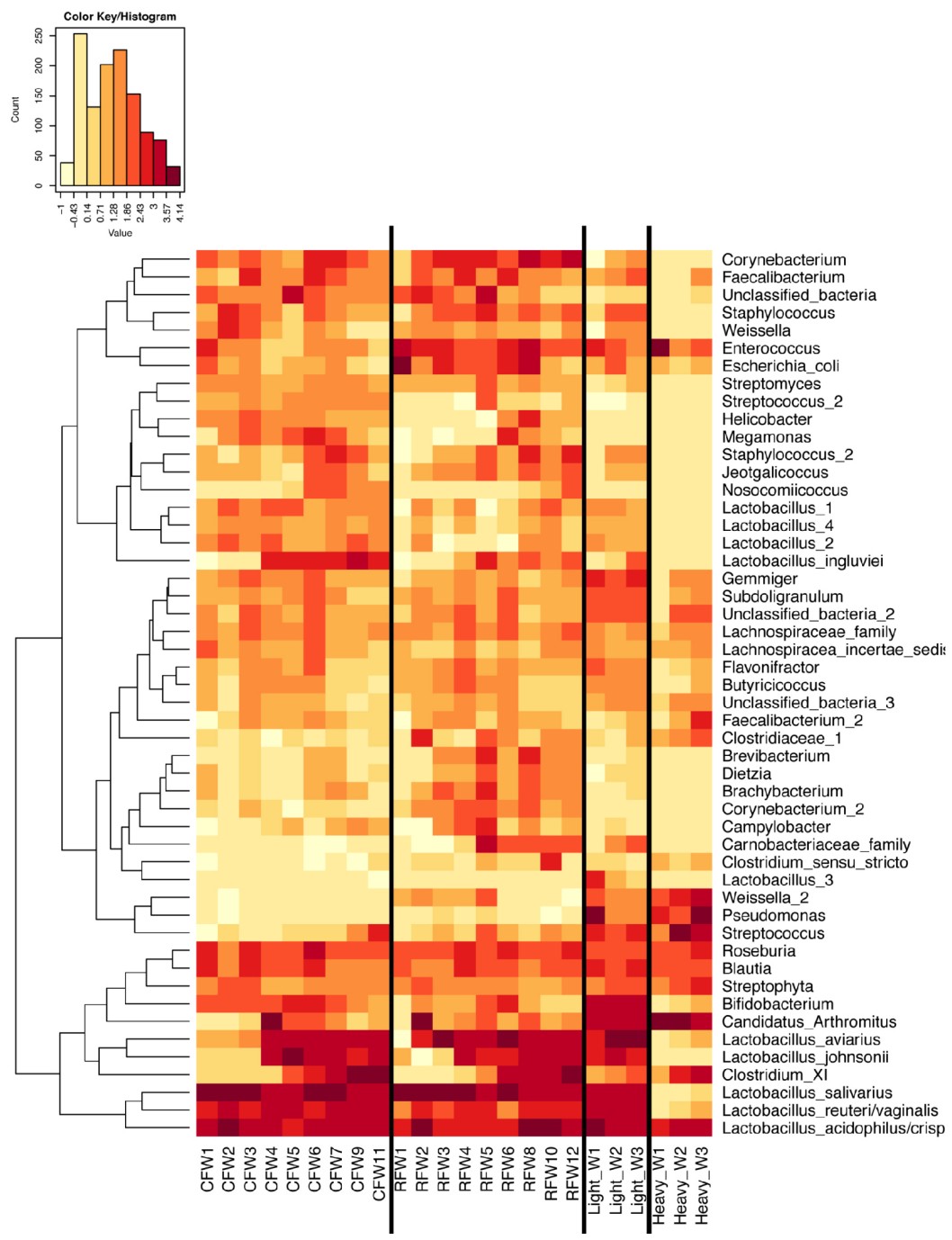

**Figure 6 Heatmap of average OTU abundance for each group studied.** Averages for multiple birds at each group and timepoint are depicted for the top 50 OTUs in this study. Heatmap is in $\log_{10}$ normalized counts.

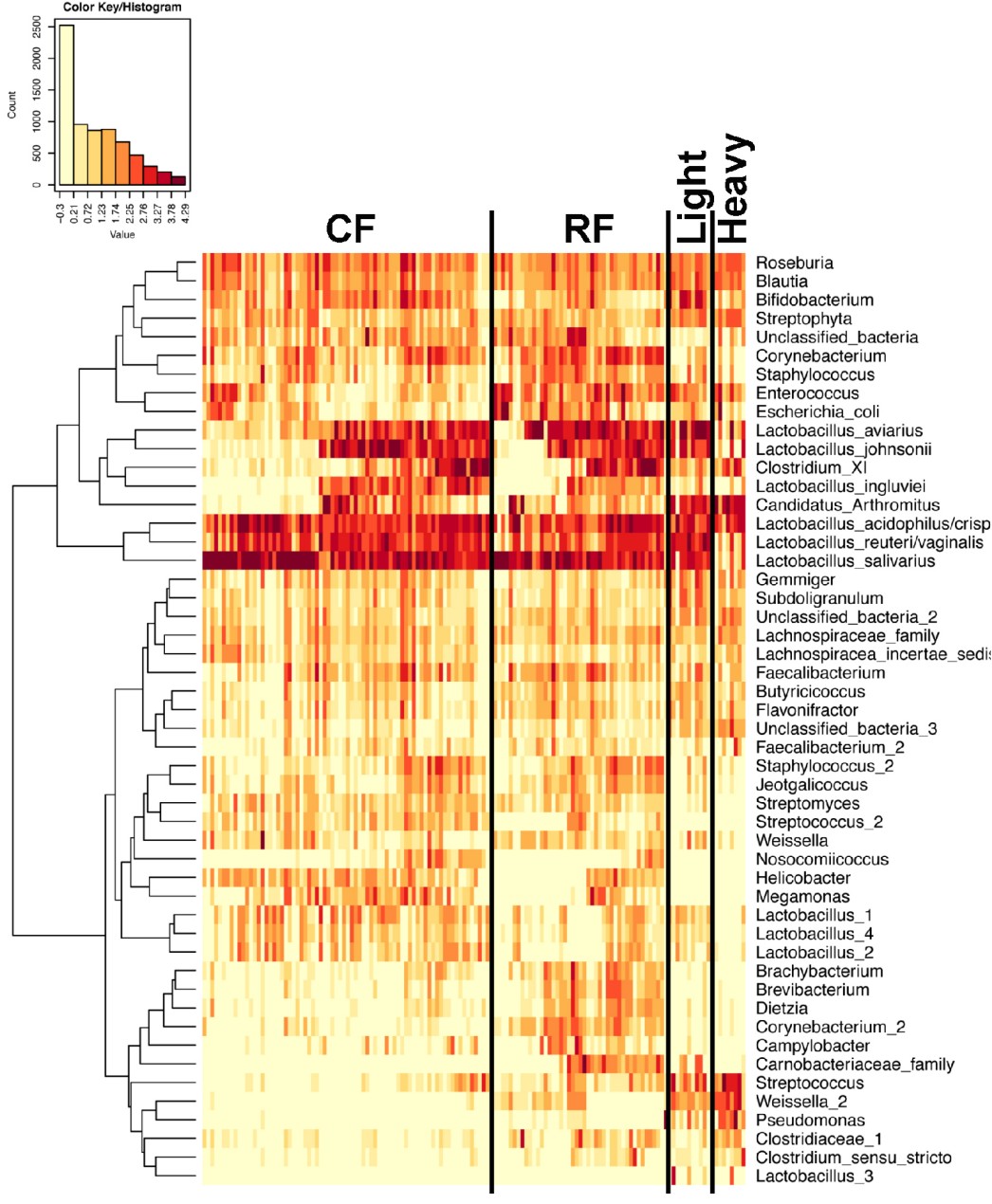

**Figure 7 Heatmap of OTU abundance in individual birds.** Normalized counts for individual birds within each group and timepoint are depicted for the top 50 OTUs in this study. Heatmap is in $\log_{10}$ normalized counts.

approximately 3 kg of weight, after which the ileum microbiome stabilized (Fig. S5). Whole bird weight also was highly correlated with intestinal weight ($R^2 = 0.83$; Fig. S6). Based upon two-way hierarchical clustering using the top 50 OTUs in the study, the Neighbor Joining cladogram illustrated that the Heavy flocks in experiment #1 clustered independently from Light flocks (Fig. 9). Overall, the clustering enabled not only clustering based upon bird age, but also clustering based upon flock source.

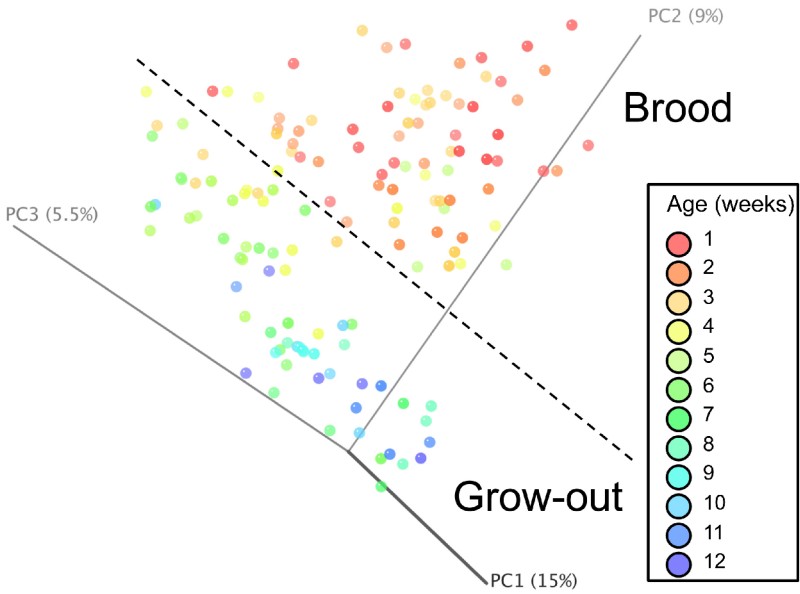

**Figure 8 Principal coordinate analysis (PCoA) of samples of differing group and timepoint.** Dashed line indicates the approximate movement of turkeys from brooder to grow-out barns. Color is based on age of turkeys in weeks.

The core microbiome was also assessed using the data from flocks CF and RF over all timepoints (Table S4). The core microbiome was defined as OTUs present in 100% of all samples at a given timepoint. From these data, a proposed model for the succession of the ileum microbiome was constructed (Fig. 10) involving the dominant bacteria that are considered core at different timepoints as the turkey ages.

## DISCUSSION

The purpose of this study was to identify differences in the bacterial microbiome in commercial turkeys of differing average flock weights, and the succession of the ileum bacterial microbiome in the growing commercial turkey. We hypothesized that differences in bacterial microbiome content could be identified that correlated with Heavy versus Light groups based upon average flock weights. In previous work, we demonstrated that age was the major driving factor in the chicken cecum bacterial microbiome when examining a single research flock, as compared to lesser community effects of in-feed antibiotic treatments (*Danzeisen et al., 2011*). Age was also a dominating factor in the bacterial microbiomes in this study, but environment also appears to play a key role in the initial stages of turkey bacterial microbiome maturation. Using RDP-based taxonomic classification, discernable patterns between Light and Heavy flocks were identified that included decreases in Bacilli and increases in unclassified bacteria later identified as Candidatus division Arthromitus. OTU-based analysis at 97% similarity enabled much greater resolution of the finer-scale microbiome changes occurring between Light and Heavy flocks, and also microbiome changes as the turkey ages. Our data suggest that

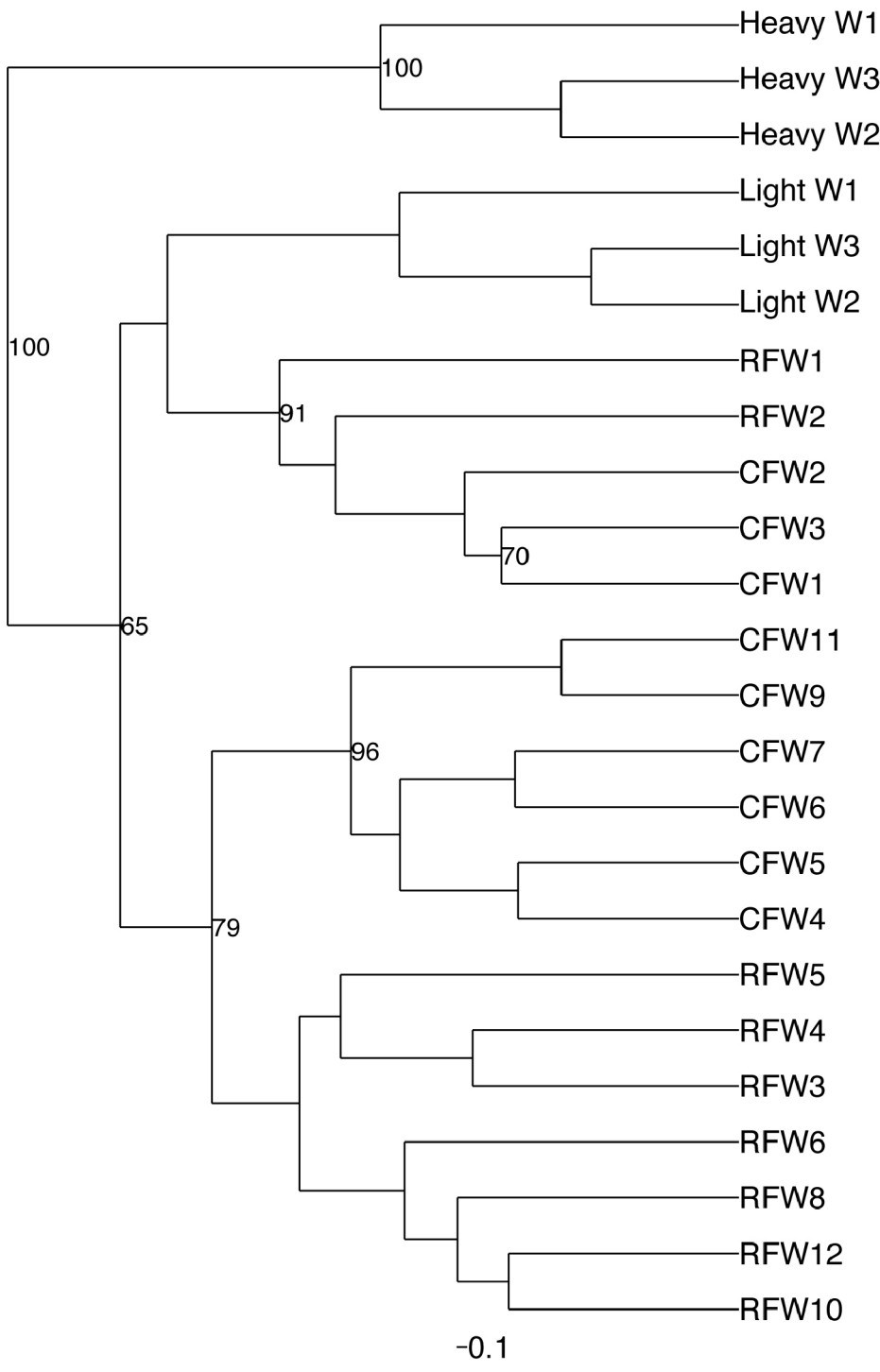

**Figure 9 Cladogram depicting relationships between groups studied.** Cladogram was generated using two-way hierarchical clustering and Neighbor-Joining algorithm. Groups are designated as "CF", commercial flock (experiment #2); "RF", research flock (experiment #2); "Heavy", flocks of heavier weights (experiment #1); and "Light", flocks of lighter weights (experiment #1).

**Weeks 1-2**
*Lactobacillus dulbrueckii* group
*Enterococcus*
*Roseburia*
*Blautia*
*+/- E. coli*

**Weeks 3-4**
*Lactobacillus salivarius*
*Lactobacillus dulbrueckii* group
*Lactobacillus aviarius*
*Lactobacillus johnsonii*
*Lactobacillus reuteri* group
*Roseburia*
*Bifidobacterim*
*Corynebacterium*
*Blautia*

**Weeks 5-7**
*Lactobacillus salivarius*
*Lactobacillus dulbrueckii* group
*Lactobacillus aviarius*
*Lactobacillus johnsonii*
*Clostridium* Group XI
*Lactobacillus reuteri* group
*Candidatus div. Arthromitus*
*Roseburia*
*Bifidobacterium*
*Corynebacterium*
*Blautia*
*Lactobacillus ingluviei*

**Figure 10  Temporal succession of bacteria in the turkey ileum.** Model is based upon OTUs identified as core in the dataset at each timepoint. Red-colored groups are present in weeks 1–2 and remain in the ileum through 7 weeks, green-colored groups emerge at 3–4 weeks, and blue-colored groups emerge at 5–7 weeks.

certain bacterial taxa can be identified that may be important in the early development of the turkey small intestine.

Some of the predominant OTUs identified in this study (with similarity to Candidatus division Arthromitus, *L. aviarius*, and *L. salivarius*) have also been identified as predominant species in the chicken mucosal microbiota (*Gong et al., 2007*). While *Lactobacillus* species are commonly used in commercial agricultural probiotics and are considered to be positively correlated with gut health, we observed an inverse correlation between bird performance and *Lactobacillus* species abundance when comparing Light versus Heavy flocks. There is precedence for this in a previous study involving broiler chickens, where it was found that *L. salivarius* and *L. aviarius* were associated with decreased bird performance (*Torok et al., 2011*). These and other studies therefore suggest that an excess abundance of certain *Lactobacillus* species in the avian small intestine may actually be indicative of decreased or slower bird development, and this could in fact be due to displacement with other bacteria such as Candidatus division Arthromitus and *Clostridium* group XI organisms that are critical for gut microbiome development (*Danzeisen et al., 2011*).

Segmented filamentous bacteria, or SFBs, are known to be indigenous members of developing microbiota in the animal small intestine. These bacteria are visible in the ileum of turkey poults in the early stages of life (*Bohorquez, Bohorquez & Ferket, 2011*). Previously, SFBs have mostly been associated with disease in poultry (*Angel et al., 1990*; *Goodwin et al., 1991*). However, these organisms are quite heterogeneous and belong to multiple

bacterial taxa (*Thompson, Mikaelyan & Brune, 2013*). Therefore, SFBs from different bacterial taxa may exert different effects on the poultry gastrointestinal tract. Some SFBs belong to the candidate taxa known as Candidatus division Arthromitus, more recently referred to as Candidatus Savagella (*Snel et al., 1995*; *Thompson, Mikaelyan & Brune, 2013*). Phylogenetically speaking, these organisms form a discrete and distant lineage within the family Clostridiaceae, clustering within Clostridia Cluster I but divergent from other organisms within this cluster (*Prakash et al., 2011*). Candidatus division Arthromitus organisms are Gram-positive, spore-forming bacteria that are of long filamentous form consisting of segmented structures. Candidatus division Arthromitus organisms are unique from other Clostridia in that they have reduced genomes and are thus highly dependent on their host for many metabolic functions, including synthesis of amino acids and purines and pyrimidines. Because of these dependencies, they have not yet been cultured. Interestingly, Candidatus division Arthromitus organisms have been previously associated with early development of the innate immune system in mice and have been positively associated with gut development in other animal species (*Prakash et al., 2011*). Therefore, whether directly or indirectly, our data suggest that Candidatus division Arthromitus-like organisms play a positive role in turkey gut development and weight gain.

While we found a clear correlation between the presence of Candidatus division Arthromitus-like organisms and Heavy flock status, it is important to note that the sequence abundance of this OTU was among the most variable identified based upon standard deviations. That is, in individual birds the presence or absence of this organism was highly unpredictable, although the average abundance of this organism using a pooled timepoint approach showed clear trends. This may underscore the reality that substantial inconsistencies exist between poults entering the brood environment that may be contributing to their variable temporal microbiome succession. Future studies are essential that address not only the bacterial microbiome in the turkey gastrointestinal tract, but also the bird immune status that corresponds with these microbiota changes.

In addition to Candidatus division Arthromitus, several other OTUs were identified that were associated with Heavy flocks and/or associated with temporal shifts in the turkey ileum microbiome. *L. aviarius* is a Gram-positive, non-spore-forming strict anaerobe that was first isolated and identified from the chicken (*Fujisawa et al., 1984*). *L. aviarius* was found to be suppressed upon *Clostridium perfringens* challenge in broiler chickens (*Feng et al., 2010*), and was also suppressed following salinomycin treatment in broilers (*Czerwinski et al., 2012*). Therefore, the emergence of *L. avaiarius* in the small intestine appears to be significant in terms of bird gut microbiota development and warrants further study.

Another OTU that was a clear marker of ileum microbiome succession in this study was 100% similar to *C. bartlettii*. This bacteria belongs to group XI *Clostridium* (*Song et al., 2004*), which is a heterogeneous group of *Clostridium* that also includes *Eubacterium* and *Peptostreptococcus* species. *C. bartlettii* is poorly described in the literature, but it has been associated with a high ability to ferment aromatic amino acids in the gut (*Russell et al., 2013*). Since *L. aviarius* and *C. bartlettii* are strict anaerobes, it is also possible that the

gastrointestinal environment dictates the ability of these organisms to colonize and expand in numbers. Whatever the case, these bacteria appear to be good markers of ileum bacterial microbiome development in the turkey.

When examining the succession of the ileum bacterial microbiome in commercial turkeys, there was a clear shift in the microbiome that occurred in all flocks examined. The timing of this shift varied between flocks examined, and in general the shift occurred earlier in research flocks than it did in commercial flocks. Clustering analysis indicated that the largest differences in the timing of this shift occurred in the Heavy research flocks in experiment #1. This is easily explained by the nature of these research flocks, which were pen-based studies, compared to all other flocks that were either actual commercial flocks or research flocks representing commercial conditions. While the timing of bacterial microbiome succession varied depending on flock studied, the succession of microbes was clear (Fig. 10) and a predictable core microbiome was established that was present in 100% of the birds examined at each timepoint (Table S4). Our model for the succession of bacteria in the turkey ileum, based upon this study, indicates that the core predictable bacterial microbiome in the turkey involves early colonization with Gammaproteobacteria at high variability (particularly *E. coli*). At the same time, consistent and dominant colonization occurs with *L. delbrueckii* group organisms, *L. salivarius*, and *L. reuteri/vaginalis*, along with colonization of lower abundance organisms including *Blautia* and *Roseburia* species. A shift occurs at weeks 3–4 of age when some birds are colonized heavily but variably by Candidatus division Arthromitus, which is accompanied by consistent colonization by *L. aviarius* and *L. johnsonii*, along with colonization of lower abundance organisms including *Bifidobacterium* and *Corynebacterium* species. At weeks 5–7 of age, *Clostridium* group XI organisms and *L. ingluviei* begin to consistently colonize the ileum. The overall timing of the key shifts appears to occur earlier in research flocks as compared to commercial flocks. The reasons for this observation are currently unknown. We can at least rule out poult source and diet, since the birds in this study came from the same hatchery and were fed similar diets. Other possible mitigating factors include management approaches and overall bird immune status, and these are variables requiring further examination.

There were some limitations to this study. First, samples were pooled for each flock and timepoint in experiment #1, limiting our ability to assess individual-to-individual variation within a given flock when comparing Light versus Heavy flocks. This was better addressed in experiment #2 addressing individual-to-individual turkey variation. Another limitation is that we performed sequencing on a small portion of the 16S rRNA gene, so the data obtained here are not indicative of functional aspects of the turkey small intestinal microbiota. Furthermore, we have a high level of confidence that the Candidatus Arthromitus-like organisms are SFBs associated with the ileum mucosa through DNA sequencing and light microscopy, but additional tools such as *in situ* hybridization would be required to confirm these observations. Furthermore, this work did not address viable counts of identified bacteria so it only measured the presence or absence of DNA representing these bacteria.

## CONCLUSIONS

This study demonstrates that the succession of bacteria in the turkey ileum proceeds in a predictable manner. The timing of bacterial succession appears to be dependent on multiple factors, at least including incoming poult consistency and commercial flock environment. From this work, several candidate markers of turkey bacterial microbiome development have been identified that will aid in future efforts aimed at modulating the microbiome to improve turkey gut health and overall bird growth and development.

## ACKNOWLEDGEMENTS

We wish to thank the independent turkey growers in Minnesota and North Dakota that graciously participated in this study. Bioinformatics was enabled through support from the Minnesota Supercomputing Institute. We thank Dr. Richard Isaacson and Dr. Srinand Sreevatsan at the University of Minnesota for thoughtful and enlightening discussions.

### Funding

This work was funded by the Minnesota Turkey Research and Promotion Council. The funder had no role in study design, data collection and analysis, decision to publish, or preparation of the manuscript.

### Grant Disclosures

The following grant information was disclosed by the authors:
Minnesota Turkey Research and Promotion Council.

### Competing Interests

Brian McComb is an employee of Willmar Poultry Company.

### Author Contributions

- Jessica L. Danzeisen performed the experiments, analyzed the data, wrote the paper.
- Alamanda J. Calvert performed the experiments, wrote the paper.
- Sally L. Noll and Catherine M. Logue conceived and designed the experiments, contributed reagents/materials/analysis tools, wrote the paper.
- Brian McComb performed the experiments, animal sampling.
- Julie S. Sherwood performed the experiments.
- Timothy J. Johnson conceived and designed the experiments, analyzed the data, contributed reagents/materials/analysis tools, wrote the paper.

### Animal Ethics

The following information was supplied relating to ethical approvals (i.e., approving body and any reference numbers):

Institutional Animal Care and Use Committee, University of Minnesota under protocol 1012B93592.

## DNA Deposition

The following information was supplied regarding the deposition of DNA sequences:

The data for this project is publicly available at http://metagenomics.anl.gov/linkin.cgi?project=3550.

## Data Deposition

The following information was supplied regarding the deposition of related data:

Figshare: http://figshare.com/authors/Timothy_Johnson/498775.

## Supplemental Information

Supplemental information for this article can be found online at http://dx.doi.org/10.7717/peerj.237.

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
