# Peer review of "Succession of the turkey gastrointestinal bacterial microbiome related to weight gain"

_PeerJ, doi:10.7717/peerj.237_

## Round 0.1 · original submission · Major Revisions

As you can see from the individual comments, both reviewers found merit in your study, which uses Next Gen sequencing of 16S rRNA gene fragments to assess potential differences in intestinal microbiota composition turkeys with different weights.
Especially, please review statistical analyses with respect to correction for multiple testing.
Also, please note that the abbreviation rDNA should be avoided, as there is no such thing as ribosomal DNA, and use rRNA gene instead.
Also, did you consider FISH to confirm that the observed mophologies indeed correspond to SFBs?

Reviewer 1 ·

Basic reporting

The introduction discusses the problem of LTS in Minnesota, is this only a MN problem?

Experimental design

Partial 16S gene sequences are not adequate to make species level assignments. This study compares relative abundances, so use of “enrichment” or “depletion” in one group or another seems misleading. The statistical analysis is not clear. The high variability in the data is noted through the manuscript, but the statistical comparisons seem to be made with no correction for multiple comparisons or false discovery. It would be useful to have a table or normalized counts to each genus so that the reader has an idea of the variability and trends.

Validity of the findings

This manuscript describes the bacterial taxonomic shifts in the turkey intestine with “heavy” and “light” flocks. This study is descriptive, uses pooled samples, and doesn’t establish if these changes are meaningful, or address questions whether these differences result from the health status of the bird, rather than the cause of their relative weight gain.

Additional comments

Line 103-104: How were sequences screened (program?) what program was used for chimera check?
Line 114: Correct citation is White et al., 2009.
Line 126: Couldn’t proteobacterial bloom reflect early succession variability?
Line 136: It is unlikely that there is Shigella in the turkey gut.
Line 185: There is no discussion about how light microscopy was used to evaluate SFB in the methods.
Line 192: microbiome should be changed to membership.
Line 230: There was no mention of figure 9 in the results, or how it was generated in the methods.

·

Basic reporting

Meet criteria

Experimental design

Meet criteria

Validity of the findings

Meet criteria

Additional comments

Using 16S rDNA-based pyrosequencing in conjunction with OTU-based analysis, the investigators observed interesting findings of turkey ileal microbiome, which may shed light on the mechsnism of Light Turkey Syndrome identified in Minnesota trurkey flocks in the past 5 years. Specifically, microbiome analysis suggested that Candidatus Arthromitus may play a beneficial role in body weight gain while high-level of Lactobacillus species may be detrimental on growth performance. As discussed in this manuscript, large scale microbiome snalaysis as well as validation work (for Candidatus Arthtromitus) are needed in future studies. The quality of this manuscript could be further improved by addressing following minor concerns:

1. Most information in the 'Conclusion' section (lines 192-250) appears to be discussion. Please move these to the 'Results and Discussion'. In addition, draft a one paragraph concise conclusion in the 'Conclusion' section.

2. The sentence in lines 185-186 (Light microscopy was used to validate that sequences with similarity to Candidatus Arthromitus appeared as segmented filamentous bacteria) is confusing and should be revised to improve clarity. In addition, this reviewer assumed that the light microscopy picture is only a putative, Candidatus Arthromitus-like organism (filamentous bacterium); the authors did not have direct evidence showing this filamentous bacterium is Candidatus Arthromitus. This issue should be better clarified in the manuscript.

---

## Round 0.2 · Minor Revisions

There are a few issues you might want to address.

In the abstract, it says barlettii, where it should be bartlettii

l.99: it says pyrosequencing, but it is not. Illumina seq is not based on pyrosequencing

l.157: Please don't use "eubacterial", but "bacterial" instead. Eubacteria is an old term from when one still distinguished between eu- and archaebacteria

l.160: phylogenetic

For the rest, please check the manuscript again carefully for additional typographic mistakes, which can just be identified using the corresponding function in Word.
Also, please make sure that all taxonomic names of microorganisms are given in italics (e.g. not the case in l.307/8

·

Basic reporting

No comments

Experimental design

No comments

Validity of the findings

No comments

---

## Round 0.3 · accepted · Accept

You have responded comprehensively to all remaining issues, even though you unfortunately didn't provide a marked-up copy with tracked changes, which made it difficult to verify the requested modifications.